# Detection of genes with differential expression dispersion unravels the role of autophagy in cancer progression

**Christophe Le Priol** [1] *, **Chloé-Agathe Azencott** [2,3,4], **Xavier Gidrol** [1] *

**1** Univ. Grenoble Alpes, INSERM, CEA-IRIG, Biomics, Grenoble, France, **2** Center for Computational Biology, Mines ParisTech, PSL Research University, Paris, France, **3** Institut Curie, Paris, France, **4** INSERM U900, Paris, France

\* chris.lepriol@gmail.com (CLP); xavier.gidrol@cea.fr (XG)

**Data Availability Statement:** The source code and data used to produce the results and analyses presented in this manuscript are available at https://github.com/lepriolc/RNAseq_differential_

## Abstract

The majority of gene expression studies focus on the search for genes whose mean expression is different between two or more populations of samples in the so-called "differential expression analysis" approach. However, a difference in variance in gene expression may also be biologically and physiologically relevant. In the classical statistical model used to analyze RNA-sequencing (RNA-seq) data, the dispersion, which defines the variance, is only considered as a parameter to be estimated prior to identifying a difference in mean expression between conditions of interest. Here, we propose to evaluate four recently published methods, which detect differences in both the mean and dispersion in RNA-seq data. We thoroughly investigated the performance of these methods on simulated datasets and characterized parameter settings to reliably detect genes with a differential expression dispersion. We applied these methods to The Cancer Genome Atlas datasets. Interestingly, among the genes with an increased expression dispersion in tumors and without a change in mean expression, we identified some key cellular functions, most of which were related to catabolism and were overrepresented in most of the analyzed cancers. In particular, our results highlight autophagy, whose role in cancerogenesis is context-dependent, illustrating the potential of the differential dispersion approach to gain new insights into biological processes and to discover new biomarkers.

## Author summary

Gene expression is the process by which genetic information is translated into functional molecules. Transcription is the first step of this process, consisting of synthesizing messenger RNAs. During recent decades, genome-wide transcriptional profiling technologies have made it possible to assess the expression levels of thousands of genes in parallel in a variety of biological contexts. In statistical analyses, the expression of a gene is estimated by counting sequencing reads over a set of samples and is defined by two dimensions: mean and variance. The overwhelming majority of gene expression studies focus on identifying genes whose mean expression significantly changes when comparing samples of

dispersion and we have archived our code on Zenodo (DOI: 10.5281/zenodo.7464716).

**Funding:** This work was supported by the Commissariat à l'Energie Atomique et aux Energies Alternatives: https://www.cea.fr/ (CLP, XG), the Université Grenoble Alpes: https://www.univ-grenoble-alpes.fr/ (CLP), MINES ParisTech: https://mines-paristech.eu/ (CAA) and the Institut national de la santé et de la recherche médicale: https://www.inserm.fr/. The funders had no role in study design, data collection and analysis, decision to publish, or preparation of the manuscript.

**Competing interests:** The authors have declared that no competing interests exist.

different conditions of interest to gain knowledge of biological processes. In this classical approach, the variance is usually considered only as a noise parameter to be estimated before assessing the mean expression. However, finely estimating the variance of expression may be biologically relevant since a modification of this parameter may reflect a change in gene expression regulation. Here, we propose to evaluate the performance of statistical methods that identify such differentially variant genes. We highlighted the potential of this approach by analyzing cancer datasets, thus identifying key cellular functions in tumor progression.

## Introduction

### Variability in gene expression in cancer

Genome-wide transcriptional profiling technologies have made it possible to assess the level of expression of thousands of genes in parallel in a variety of biological contexts [1]. Cells or organs are commonly characterized by the mean expression of some key genes [2]. As a consequence, phenotypes are defined to be driven by a change in the mean expression of some genes between sets of samples that represent conditions of biological interest, *e.g.* diseased and healthy status [3]. Several methods have thus been developed to identify these genes, called "differentially expressed" (DE) genes. This has led to numerous insights into a variety of biological processes [4, 5]. Differentially expressed genes may also serve as biomarkers [6]. In this type of analysis, the variability is often reduced to "noise" that one must remove. Consequently, variability is considered to be a parameter that must be estimated prior to searching for a difference in mean expression. However, in the same manner that the level of expression of a gene has biological meaning, the variability of its expression is another trait of its biological function [7, 8]. For example, low gene expression variability defines housekeeping genes [9, 10].

The fluctuations in gene expression may indeed be driven by a variety of intrinsic sources, *e.g.* the stochastic nature of gene transcription [11], the cell cycle [12], stochastic regulation [13], chromatin modification [14] or mRNA degradation [15], as well as extrinsic causes, which refer to all environmental perturbations [16, 17]. In cancer, the overall increase in gene expression variability [18] is a way for tumors to resist therapy [19, 20]. In addition, it may reveal a lack of precision in gene expression, which tends to be highly controlled in healthy conditions [21, 22]. For these reasons, variability is a relevant trait in gene expression to gain better knowledge of cancer development.

### Estimation of gene expression variability

The terms "variability" and "variation" are often used to describe how much the expression of a gene fluctuates when comparing different samples. These terms may be confusing when analyzing samples from different biological conditions, since they are commonly used to refer to a change of mean expression between conditions. In addition, they are not statistical terms and should therefore be replaced by the metric used to estimate the variability in the analyzed data. A myriad of measures may be used to estimate gene expression variability, *e.g.* the variance, the standard deviation, the coefficient of variation (CV), the median absolute deviation, the expression variability [10], the Shannon entropy [23] or the expression change [21].

Genes having a difference of variance in expression between biological conditions of interest are called "differentially variant" (DV) genes and are identified using basic statistical

approaches: F-test to compare variances [8, 24], Wilcoxon rank-sum test to compare CVs [25, 26], differences of entropy tests [23] or comparison of CV distributions to random distributions using Wilcoxon's signed rank test [27]. A few studies have focused on analyzing gene expression variability and identified genes with differential variance in different biological contexts: cancer progression [8, 24], neurologic diseases such as Parkinson's disease and schizophrenia [25, 28] or between cell populations in development [26]. Most of these studies used microarrays and log-transformed the expression data prior to measuring gene expression variability. This transformation affects the mean-variance relationship [29] and therefore appears to be suboptimal for estimating gene expression variability.

High-throughput sequencing of the transcriptome (RNA-seq) has become the gold-standard technology to estimate genome-wise gene expression [30]. Contrary to microarray data, RNA-seq count data are integer values, which makes log-transformation, usually performed with microarray data, not appropriate for this type of data [31]. Therefore, dedicated methods based on discrete probability distributions were developed to analyze these data [32]. The negative binomial (NB) distribution has become the ubiquitous distribution to model RNA-seq read count data by providing the best fit for the extra-variance commonly observed in datasets composed of biological replicates [33]. In this model, the random variable describing the count of reads mapped to gene $i$ in sample $j$ is denoted as $Y_{ij} \sim \mathcal{NB}(\mu_{ij}, \phi_i)$, where $\mu_{ij}$ is the expected value and $\phi_i$ is the dispersion parameter. The variance is given by $\mathrm{Var}(Y_{ij}) = \mu_{ij} + \phi_i \mu_{ij}^2$. Analyzing the variance independently with respect to the mean expression can therefore be achieved by analyzing the dispersion parameter $\phi_i$.

In the classical RNA-seq data analysis workflow, differential expression detection methods based on the NB distribution consider the dispersion as a noise parameter to be estimated prior to identifying a difference of mean expression [34]. The generally low sample sizes of RNA-seq datasets at the time when the first versions of these methods were published made per-gene dispersion estimation unreliable. In addition, the very high number of genes made estimation difficult. Thus, Robinson *et al.* proposed an accurate shared estimator based on the expression of sets of genes across all samples, independent of biological condition [33]. Per-gene estimators were then shrunk towards this shared estimator using different levels of shrinkage [35–38]. Aggregating all the samples that compose the dataset implies that no difference of dispersion in the expression of genes between the conditions of interest can be modeled, which is not biologically realistic.

## Identification of differently dispersed genes

Recently, four methods based on the NB distribution, MDSeq [39], DiPhiSeq [40], the analysis proposed by de Jong *et al.* [41] and DiffDist [42], have been introduced to identify differences in both mean and dispersion in RNA-seq data within the same statistical framework.

MDSeq [39] extends the use of a generalized linear model (GLM) to identify both mean and dispersion differences by reparameterizing the NB distribution with a linear mean-variance relationship: $\mathrm{Var}(Y_{ij}) = \phi_{ij} \mu_{ij}$. Since the NB distribution with a varying dispersion parameter does not belong to the exponential family, the usual closed-form estimates for the GLM parameters cannot be used. Instead, the minimization of the log-likelihood of the model is formulated as an optimization problem with linear inequality constraints that can be solved using an adaptive barrier algorithm combined with the BFGS algorithm. Wald tests are then performed to identify differential expression mean and dispersion.

DiPhiSeq [40] implements a GLM allowing a single explanatory variable, the comparison of interest, but, unlike the classical differential expression methods, estimates the dispersion for each gene and for the two compared conditions. Because of the high sensitivity of the

likelihood ratio test to outliers, the authors of DiPhiSeq used robust M-estimators to estimate both the mean and the dispersion in both conditions. In this approach, the Tukey's biweight function is used as the function to minimize. Differences in the mean and dispersion are finally compared to a standard distribution under the null hypothesis of no difference and p-values for differential expression and differential dispersion are deduced.

De Jong *et al.* [41] proposed to take benefit of a generalized additive model for location, scale and shape (GAMLSS), an extension of a GLM which allows the estimation of differences of dispersion between groups of samples. In this approach, statistics and p-values are obtained thanks to a likelihood ratio test between a GLM, which stands for a reduced model for the dispersion, and the full GAMLSS model. Hereafter, we denote this approach as GAMLSS.

DiffDist [42] implements a hierarchical Bayesian model which estimates mean and dispersion parameters based on log-normal priors whose location and scale parameters are modeled by normal and gamma hyperprior distributions, respectively. Posterior samples of the mean and dispersion parameters are obtained using an adpative Markov chain Monte Carlo algorithm. The posterior samples of log-fold change between two groups of samples are then deduced and tail probabilities are obtained using highest posterior density intervals to estimate the probability of no difference in parameter values between groups.

## Objectives

The performances of methods identifying differences in mean expression in the so-called "differential expression analysis" using RNA-seq data have been extensively studied [43–46]. The large amount of publicly available RNA-seq data opens new perspectives for researchers in the search for genes whose expression exhibits a difference of dispersion between samples from different conditions and a new space for the discovery of biomarkers. To that end we first evaluate the performances of four NB-based methods, MDSeq, DiPhiSeq, GAMLSS and DiffDist, and a statistical test of difference of variances, Levene's test [47], to identify differentially dispersed (DD) genes using simulated RNA-seq datasets. Based on our simulation study results, we reliably applied these methods to The Cancer Genome Atlas datasets and identified DD genes that could not be identified by the classical differential expression analysis. We showed that these genes may serve to better understand tumor progression and thus have demonstrated the potential of the differential dispersion approach in RNA-seq studies.

## Results

### Evaluation of differential dispersion detection performances

**Differential dispersion detection for genes with unconstrained differences in mean expression.** We simulated RNA-seq datasets to evaluate the performances of MDSeq, DiPhiSeq, GAMLSS and DiffDist to identify differential dispersion between two sets of samples of equal size that represent two conditions of interest. We also included Levene's test to identify differences of variances after $\log_2$-transformation of the data. Differences in the mean and dispersion between the two sets of samples were introduced and defined for DE and DD genes, respectively (see the Methods section for more details). Overall, GAMLSS and DiffDist outperform all the other evaluated methods for all samples sizes (green boxplots in Fig 1).

It is noteworthy that Levene's test performs only a bit worse than DiPhiSeq and better than MDSeq. The lower area under the ROC curve (AUC) and sensitivity values obtained with MDSeq may be explained by the difference in false discovery rate (FDR) controlling procedures: the Benjamini-Yekutieli procedure for MDSeq and the Benjamini-Hochberg procedure for all the other methods, as recommended by the respective authors of the different methods. The Benjamini-Yekutieli procedure is indeed more conservative than the former [48]. We

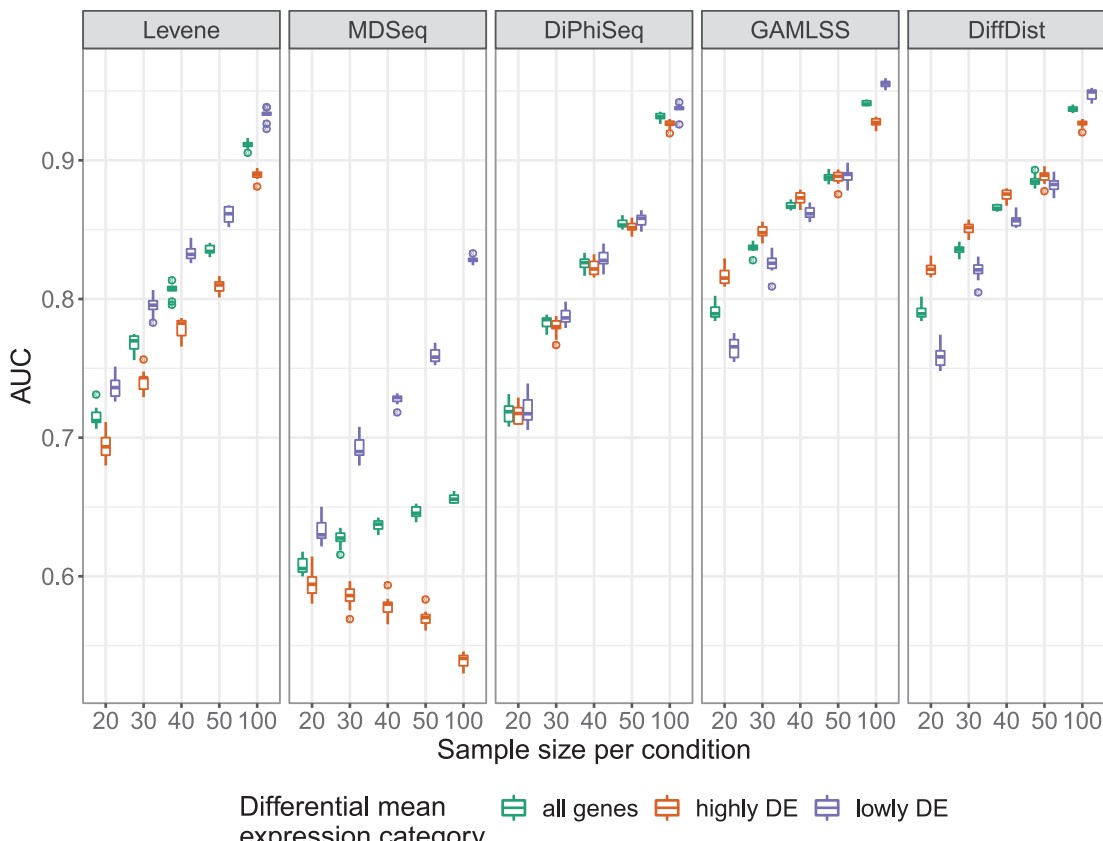

**Fig 1. Ability to identify differentially dispersed genes.** The performances of Levene's test, MDSeq, DiPhiSeq, GAMLSS and DiffDist for differential dispersion detection in gene expression data, as measured by the area under the ROC curve (AUC), were assessed using 10 replicates of simulated datasets composed of highly and lowly differentially expressed genes between two sample populations of equal size.

note, however, that in our evaluation, the Benjamini-Hochberg procedure was unable to maintain the FDR below 0.05 when dealing with the entire set genes, but also when focusing on lowly DE genes (S1 Fig).

As expected, increasing the number of samples available per condition increases the ability to detect differential dispersion. Nevertheless, these sample sizes are much larger than those usually required to achieve similar performances in the classical differential expression analysis [43–45]. For example, 40 samples per condition are required for DiPhiSeq to obtain an AUC higher than 0.8, and sets of 50 samples are required for MDSeq to obtain an AUC close to this value among lowly DE genes, while only 5 samples may suffice to identify differences in mean expression with this performance [45]. By lowering the sample size to 20 to 30 samples per condition to reach this value of AUC, the proposition made by de Jong *et al.* of using GAMLSS and the recent development of DiffDist provided a major improvement in differential dispersion detection.

The other main result of our simulation study is the sensitivity of the performance of most of the evaluated methods to the presence of a difference in mean expression between the two compared sets of samples (orange and purple boxplots in Fig 1). A fold change in the mean sharply reduces the performance of MDSeq and, to a lesser extent, Levene's test for differential dispersion detection. The application of MDSeq to identify differential dispersion must therefore be restricted to non- or lowly DE genes. The opposite is observed with GAMLSS and

DiffDist for low sample sizes while populations of 100 samples revert this sensibility. By contrast, DiPhiSeq is not sensitive to the presence of a difference in mean expression between the two compared sets of samples.

**Differential dispersion detection for lowly DE genes.** We therefore investigated the maximum difference in means of gene expression according to the number of samples in the two compared conditions, in particular to maintain the reliability of the differential dispersion detection with MDSeq. Given the results in Fig 1, this number is expected to depend on the number of samples. Fig 2 shows the performances of Levene's test, MDSeq, DiPhiSeq, GAMLSS and DiffDist on simulated datasets stratified by the maximum tolerated mean expression fold change value and the number of samples per condition.

The trends observed with datasets with high mean fold change values are confirmed by the results obtained with simulated datasets with moderated mean fold change values. GAMLSS and DiffDist outperform again all the other evaluated methods. They are indeed much sensitive to detect DD genes than all the other evalutated methods while controlling the FDR, in particular for low sample sizes. DiPhiSeq is the more robust method with very low FDR values for all the evaluated sample sizes while being also the less sensitive method, in particular for low sample sizes. DiPhiSeq is indeed unable to detect any DD genes with fewer than 40 samples per population. However, increasing the sample size to 100 samples per population enabled DiPhiSeq to reach a sensitivity similar to the one observed with Levene's test and close to those of

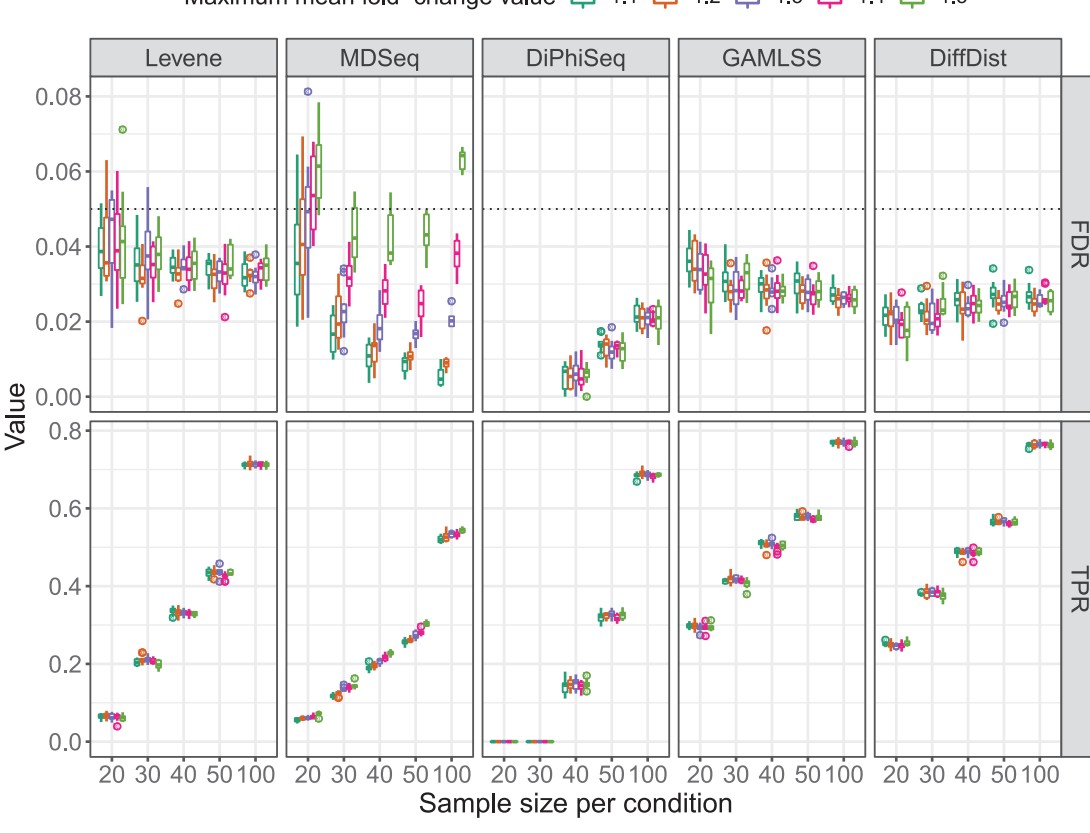

**Fig 2. Ability to detect differential dispersion for lowly differentially expressed genes.** False discovery rate (FDR) and true positive rate (TPR) of Levene's test, MDSeq, DiPhiSeq, GAMLSS and DiffDist for differential dispersion detection in simulated datasets composed of lowly differentially expressed genes between two sample populations of equal size. The performances were assessed using 10 replicates of simulated datasets per parameter setting.

GAMLSS and DiffDist. Regarding MDSeq, increasing the maximum tolerated mean fold change value increases the FDR for the detection of differential dispersion. However, the FDR remained below 0.05 for datasets composed of 30 to 50 samples per population with maximum tolerated mean fold changes up to 1.5. When only 20 samples per population are available, the maximum tolerated mean fold change value must be at most 1.3 to keep the FDR below 0.05. Similarly to simulated datasets with high values of mean fold change, we evaluated the Benjamini-Hochberg procedure and observed that it was again unable to maintain the FDR below 0.05 for almost all datasets, reaching values above 0.1 for large mean fold change values (S2 Fig).

Nevertheless, we observed some differences with these simulated datasets. Contrary to datasets with high mean fold change values, the performances of Levene's test, GAMLSS and DiffDist are not affected by the presence of the low mean fold change values. Besides, Levene's test appears to be even more sensitive than MDSeq and DiPhiSeq while maintaining the FDR below 0.05 for most replicates.

The limitation in the application of MDSeq to lowly DE genes is not prohibitive since the purpose of our approach is to identify genes that would not be detected by the classical differential expression analysis or, at least, that would not appear in the top results of these analyses. Thus, in our approach, these genes represent the set of genes of primary interest among which to search for differential dispersion in expression. MDSeq provides the possibility to use specific threshold values to identify both DE and DD genes. We therefore identified that the maximum mean fold change threshold values that maintain the FDR of the differential dispersion detection below 0.05 vary from 1.15 to 1.30 depending on the sample size (S3 Fig).

We then explored how much the sets of true positive results, *i.e.* the genes simulated with a fold change of dispersion between the two conditions and identified as DD genes, of the different methods are consistent with each other. Results for populations of 50 samples are showed in Fig 3 and results for the other sample sizes are showed in S1 File.

Regarding large sample sizes, *i.e.* populations of 50 and 100 samples, most of the true DD genes are identified by all the evaluated methods. As a consequence of their higher sensitivity, GAMLSS and DiffDist are the methods which specifically identify the most DD genes either in common or separately. Hundreds of genes are indeed not detected by any of the other methods. These sets of specific DD genes are even the biggest sets for low sample sizes from 20 to 40 samples per condition (S1 File). The three other evalutated methods specifically identify only dozen of DD genes, *e.g.* from 24 for DiPhiSeq to 61 for MDSeq for populations of 50 samples. Nevertheless, the much higher sensitivity of GAMLSS and DiffDist must be mitigated by the presence of errors in the $\log_2$-fold change sign of the detected DD genes (Fig 3B). For populations of 50 samples, 5.73% and 5.76% of the DD detected genes by GAMLSS and DiffDist, respectively, have indeed incorrect signs of dispersion $\log_2$-fold change while the other methods does not make this type of mistakes (DiPhiSeq) or for very rare cases (Levene's test or MDSeq). Moreover, the proportion of erroneous $\log_2$-fold change signs increase as the sample size decreases to reach 13.27% and 14.12% of the DD detected genes by GAMLSS and DiffDist, respectively, for populations of 20 samples (S1 File). We explored into more details these genes with incorrect $\log_2$-fold change sign according to GAMLSS and DiffDist and found that these errors occur exclusively for genes with a decrease in dispersion of expression in the second condition, irrespective of the variation in mean expression between the two conditions for both methods (Fig 3C). Then, we investigated the corresponding $\log_2$-fold change signs according to the three other methods for these particular genes. Levene's test makes the same errors while MDSeq and DiPhiSeq correctly predict the sign for most of them, with higher proportions for DiPhiSeq (Fig 3D). Thus, taking advantage of the robustness of DiPhiSeq, we recommend to validate the DD genes detected by GAMLSS and DiffDist by keeping those with consistent $\log_2$-fold change signs according to DiPhiSeq.

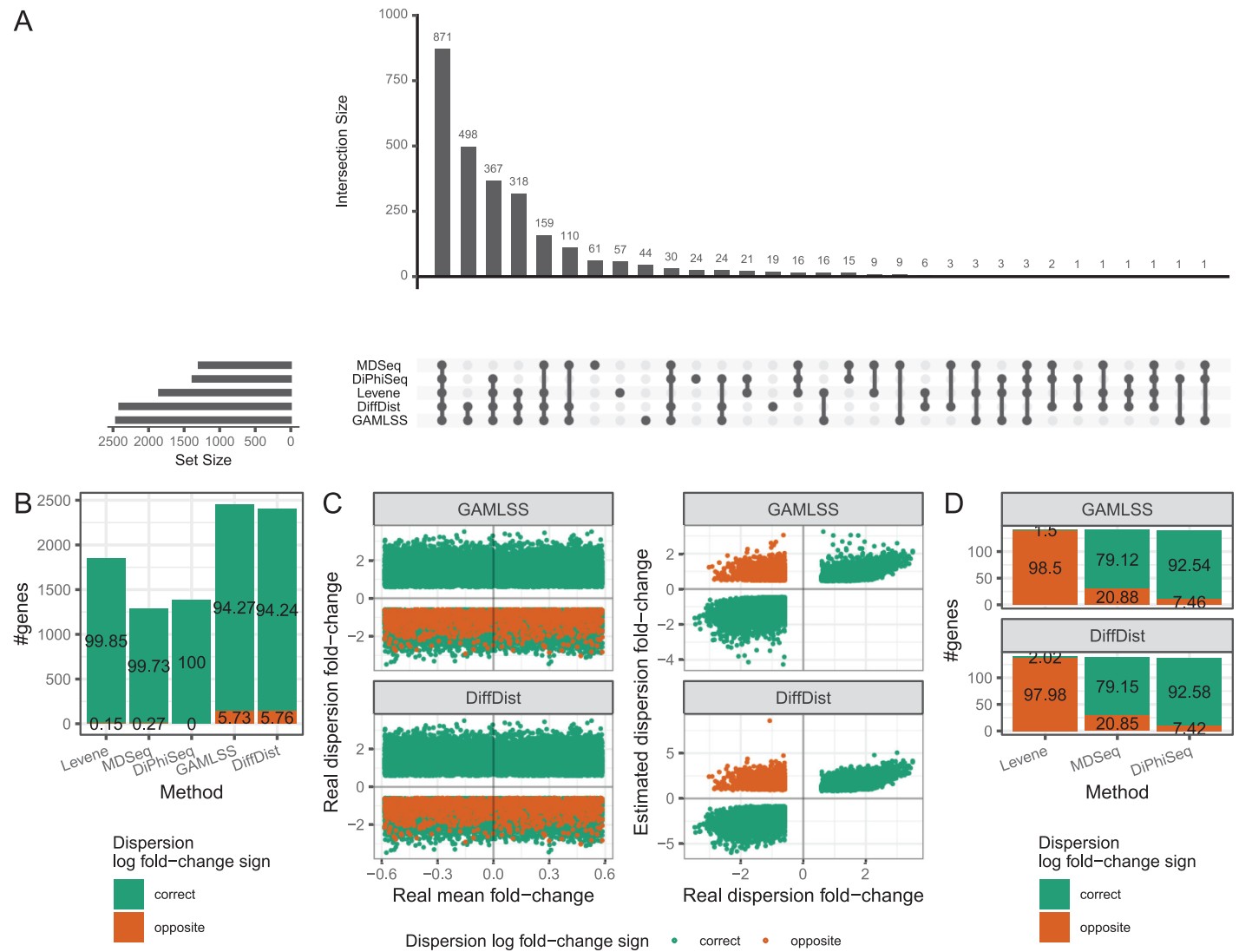

**Fig 3. Differentially dispersed genes correctly identified by the evaluated methods among lowly differentially expressed genes.** (A) Intersections of sets of differentially dispersed (DD) genes correctly identified by Levene's test, MDSeq, DiPhiSeq, GAMLSS and DiffDist. (B): Correctness of dispersion $\log_2$-fold change sign of DD genes correctly identified by the different methods. (C) Real mean and dispersion $\log_2$-fold changes and estimated dispersion $\log_2$-fold changes of DD genes correctly identified by GAMLSS and DiffDist. (D) Correctness of dispersion $\log_2$-fold change signs according to Levene's test, MDSeq and DiPhiSeq for DD genes correctly identified by GAMLSS and DiffDist with incorrect dispersion $\log_2$-fold change sign. Simulated datasets are composed of lowly differentially expressed genes with a mean fold change of expression between 1 and 1.5 between two populations of 50 samples. Values indicated at the middle of the bars are percentages of the corresponding categories of genes over the entire sets of analyzed genes. All results relate to 10 replicates of simulated datasets, *e.g.* the counts and percentages are averaged over all the replicates.

## Differential dispersion in gene expression in cancer

We applied the five evaluated methods to The Cancer Genome Atlas (TCGA) datasets [49] to identify DD genes when comparing normal and tumor samples. We used RNA-seq data from patients for whom tumor tissue and adjacent normal tissue samples were available, to limit individual variability. In agreement with the results of our simulation study, only the datasets with more than 30 samples for both conditions were analyzed, in order to maintain the FDR below 0.05 with MDSeq and to ensure sufficient power with DiPhiSeq. We list these datasets in Table 1.

**Table 1. Numbers of adjacent normal and tumor samples for the analyzed TCGA datasets.**

| Dataset | Samples | |
|---------|---------|---------|
|         | **normal** | **tumor** |
| TCGA-BRCA | 112 | 117 |
| TCGA-COAD | 41 | 46 |
| TCGA-HNSC | 43 | 43 |
| TCGA-KIRC | 72 | 72 |
| TCGA-KIRP | 31 | 31 |
| TCGA-LIHC | 50 | 50 |
| TCGA-LUAD | 57 | 67 |
| TCGA-LUSC | 49 | 49 |
| TCGA-PRAD | 52 | 54 |
| TCGA-THCA | 58 | 58 |

The samples originate from patients for whom samples of tumor tissues and adjacent normal tissues are available. Only the datasets with at least 30 samples for both conditions are analyzed: BRCA (BReast invasive CArcinoma), COAD (COlon ADenocarcinoma), HNSC (Head and Neck Squamous cell Carcinoma), KIRC (KIdney Renal Clear cell carcinoma), KIRP (KIdney Renal Papillary cell carcinoma), LIHC (LIver Hepatocellular Carcinoma), LUAD: (LUng ADenocarcinoma), LUSC (LUng Squamous cell Carcinoma), PRAD (PRostate ADenocarcinoma), and THCA (THyroid CArcinoma). For some datasets, the numbers of samples from normal and tumor tissues are different because several samples from tumors are available and are integrated in the analysis.

**Identification of DD genes among non-DE genes.**   Taking benefit of the GLM implemented in MDSeq, we identified DE genes while taking into account batch effects and identified DD genes among non-DE genes using Levene's test, MDSeq, DiPhiSeq, GAMLSS and DiffDist. We used a fold change threshold of 1 to identify both DE genes and DD genes among non-DE genes (Fig 4).

The identification of DE genes by MDSeq leaves several thousand genes among which DD genes can be searched for in any dataset. Among non-DE genes, the majority of DD genes are overdispersed in tumors (DD+). Overall, all the methods generate consistent results: some cancers are characterized by a high number of DD+ genes (breast, colon, kidney, liver and lung), while others contain much less DD genes (head and neck, prostate and thyroid).

Consistent with our simulation study, most of the DD+ genes are identified by all the five evaluated methods and GAMLSS and DiffDist are the methods that specifically identifies the most DD+ genes for most datasets, like in the kidney renal clear cell carcinoma dataset (TCGA-KIRC, Fig 5A).

Hundreds of DD+ genes are indeed only detected by these two methods (S2 File). These sets of specific genes may represent a very significant part in the overall set of DD+ genes detected by all the methods like for the prostate cancer (S2 File), highlighting the breakthrough of GAMLSS and DiffDist among the methods identifying differential dispersion in RNA-seq data. Nevertheless, the two other differential dispersion methods specifically identify significant sets of genes for some datasets, such as the kidney renal papillary cell carcinoma (TCGA-KIRP, Fig 5B) and prostate adenocarcinoma datasets (TCGA-PRAD, S2 File) for MDSeq, the thyroid carcinoma dataset (TCGA-THCA) for DiPhiSeq (S2 File) and, to a lesser extent, the head and neck squamous cell carcinoma dataset (TCGA-HNSC) for both (S2 File). These sets of genes are all the more important that they deal with the three datasets for which the less DD + genes are detected over all the analyzed datasets. Finally, Levene's test is the method which specifically identifies the less DD+ for most datasets.

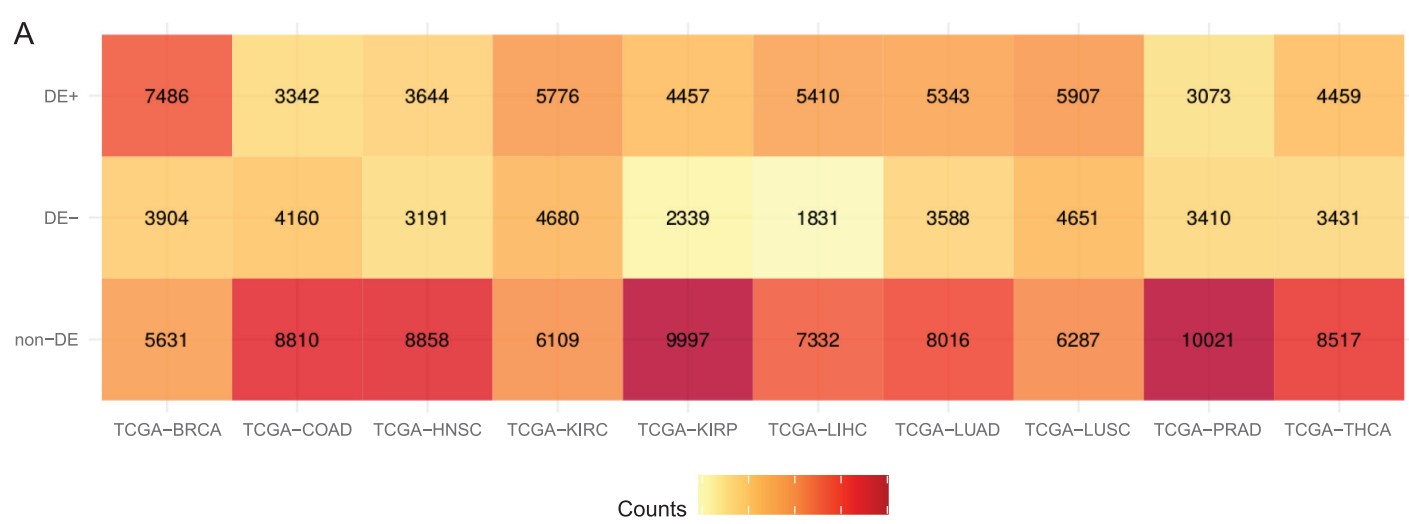

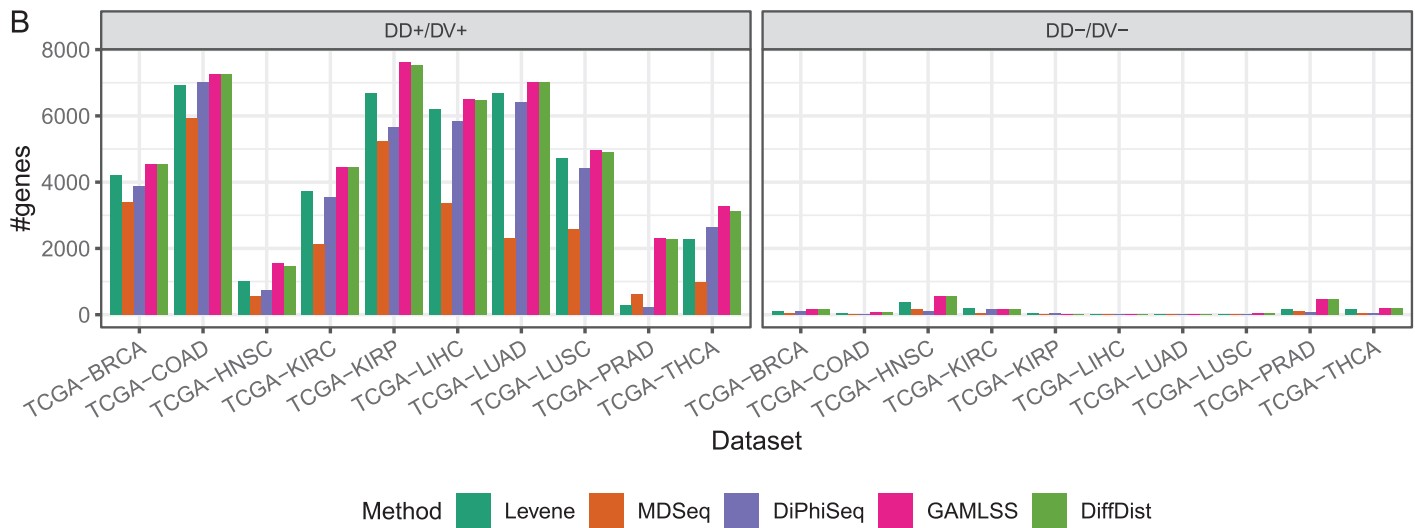

**Fig 4. Differentially dispersed genes among non-differentially expressed genes for each TCGA dataset.** (A) Number of differentially expressed (DE) genes separated between those upregulated in tumors (DE+) and those downregulated in tumors (DE-) detected by MDSeq per TCGA dataset. (B) Number of differentially dispersed (DD) genes among non-DE genes separated between those overdispersed in tumors (DD+) and those underdispersed in tumors (DD-), as detected by Levene's test, MDSeq, DiPhiSeq, GAMLSS and DiffDist, per TCGA dataset.

**Gene Ontology term enrichment analysis.** To gain biological insight, we conducted an analysis of enrichment in Gene Ontology (GO) terms among the previously identified DD + genes for each TCGA dataset. Since DD+ genes are identified with an FDR below 0.05 for all methods according to our simulation study, the entire set of DD+ genes identified by at least one method is taken into account to gain the most biological knowledge in the GO term enrichment analysis for each dataset. As we showed in our simulation study, GAMLSS and DiffDist may incorrectly estimate the sign of dispersion $\log_2$-fold change. Since this type of error is detrimental for the subsequent GO term enrichment analysis, we only retained the DD genes identified by these two methods whose dispersion $\log_2$-fold change sign is in agreement with the one predicted by DiPhiSeq (S4 Fig). We used redundancy reduction methods to ease the comparison of enriched GO terms across all the analyzed datasets (see Methods for more details). The top 40 representative terms and the p-values of their enrichment in each dataset are shown in Fig 6. The full list of enriched representative GO terms is available in S3 File.

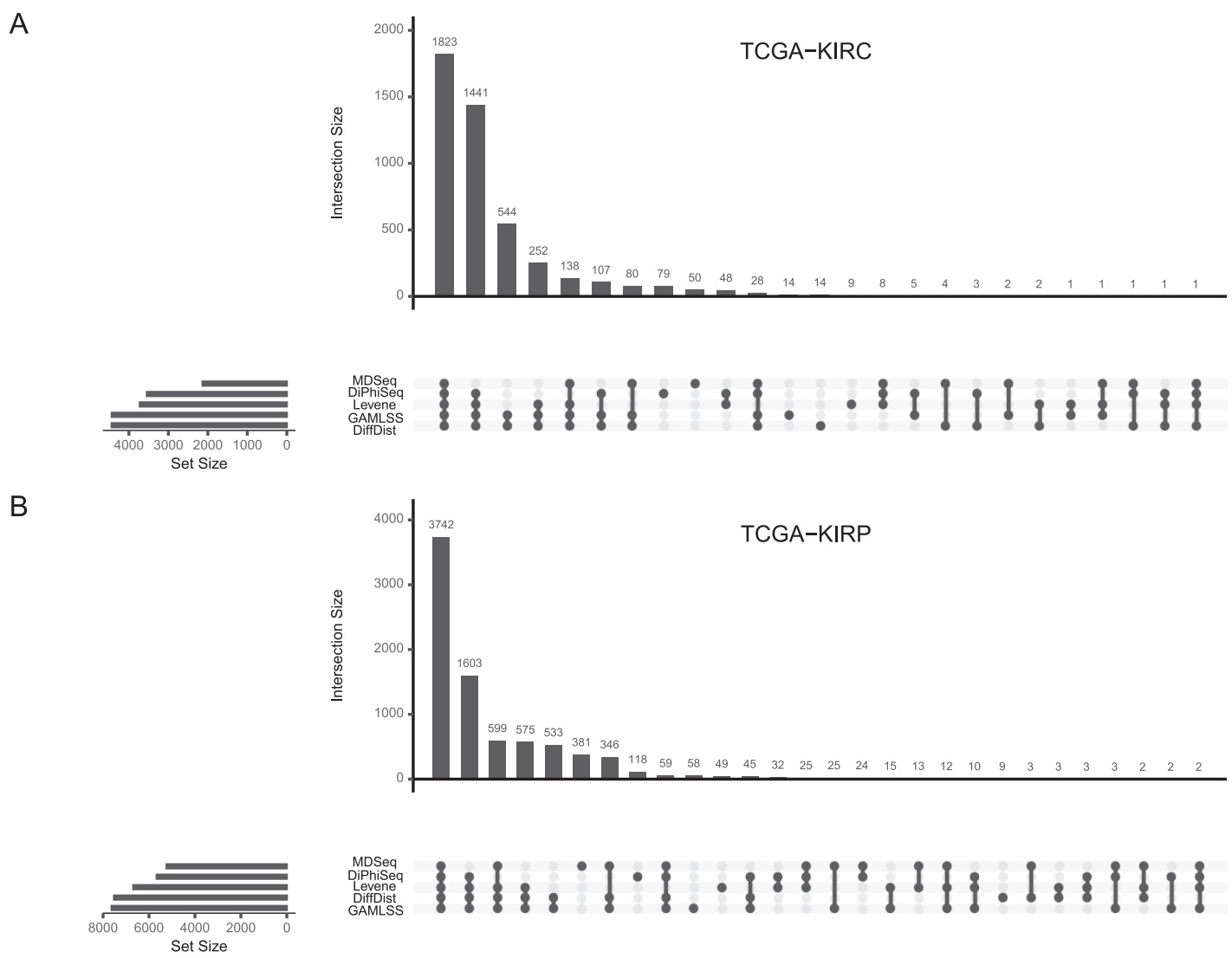

**Fig 5. Overdispersed genes in tumors identified by the evaluated methods among non-differentially expressed genes.** Intersections of sets of overdispersed genes in tumors identified by Levene's test, MDSeq, DiPhiSeq, GAMLSS and DiffDist among non-differentially expressed genes for (A) the kidney renal clear cell carcinoma dataset (TCGA-KIRC) and (B) the kidney renal papillary cell carcinoma dataset (TCGA-KIRP). Non-differentially expressed genes were identified by MDSeq.

Interestingly, among DD+ genes, the most significantly enriched GO terms were the most widespread across all the analyzed tissues and focused on some key cellular functions, such as catabolism. In contrast, GO terms that were found to be significantly enriched for only a few datasets tended to have higher p-values than the most widely enriched GO terms (S3 File). This striking result suggests some common features in tumoral development and progression, regardless of the tissue of origin, whose involved gene expression is characterized more by an increase in dispersion than by a change in the mean in tumors.

It is worth noting that many of these common biological processes are related to catabolism, *e.g.* "mRNA catabolic process", "protein targeting" or "proteasomal protein catabolic process", as previously shown by Han *et al.* [27]. In particular, several processes related to the ubiquitin-proteasome system, which is a major controller of the protein degradation process and is highly involved in cancer [50], are found among the most significant results ("protein

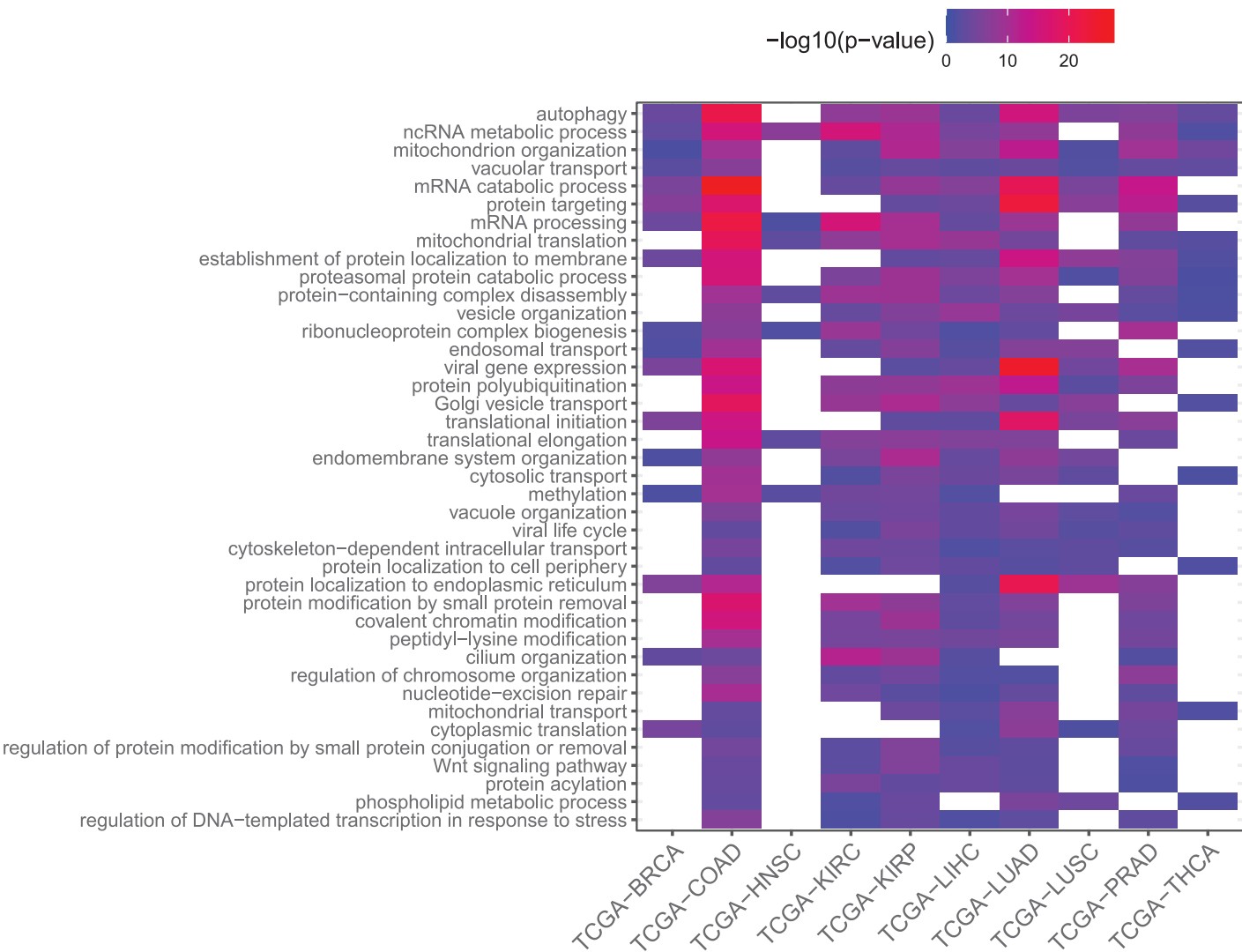

**Fig 6. Enriched GO terms among overdispersed genes in tumors identified by the evaluated methods.** Top 40 representative enriched Gene Ontology (GO) terms among overdispersed genes in tumors (DD+) among non-differentially expressed (non-DE) genes, ordered first by the number of datasets for which they are enriched (decreasing order) and second by the mean p-values of enrichment across all datasets (increasing order). Non-DE genes were identified using MDSeq, and DD+ genes were identified among non-differentially expressed genes by at least one of the evaluated methods, *i.e.* Levene's test, MDSeq, DiPhiSeq, GAMLSS and DiffDist.

polyubiquitination", "proteasomal protein catabolic process"). In contrast, no process related to anabolism was found among the most frequently enriched processes among DD+ genes, suggesting that catabolic processes are much more affected by the dysregulation of gene expression than are anabolic processes.

Autophagy was also found as the most widespread and significantly enriched biological process among DD+ genes for all the analyzed datasets. Similar to the proteasome, it is a main recycling system for biological molecules that enables cells to survive critical situations such as nutrient starvation and the degradation of damaged organelles or pathogens. In pre-malignant cells, autophagy actively acts to preserve the physiological homeostasis of multiple functions, *e. g.* elimination of mutagenic entities, decrease local inflammation, and thus aid the struggle against tumor development. In malignant cells, autophagy affects the tumor progression and the response to treatment in multiple ways, some of which act in opposition. Autophagy

desensitizes cells to programmed cell death mediated by different treatment strategies but is also involved in danger signal emission which triggers an immune response through antigen presentation. Thus, the overall effect of autophagy on tumor progression and response to treatment is context-dependent [51]. The increase in the dispersion in the expression of genes involved in autophagic processes reveals the complexity of these processes in tumor progression and may lead one to wonder whether they should be induced or, on the contrary, inhibited as a cancer treatment [52]. Some treatments indeed aim to stimulate these processes, while others aim to inhibit them [53].

## Discussion

We thoroughly assessed the performance of methods to detect differential dispersion in RNA-seq data. Based on simulated datasets, we characterized Levene's test, MDSeq, DiPhiSeq, GAMLSS and DiffDist performances and proposed recommendations to reliably apply these methods to real datasets.

### Gene expression dispersion in cancer

**Overall increase of dispersion and robustness.**   By applying the five evaluated methods to TCGA datasets, we identified an overall increase in the dispersion in the expression of many non-DE genes in tumors in comparison with normal tissues. Also analyzing TCGA datasets, Han *et al.* have already revealed an increase in the coefficient of variation of gene expression in tumors of breast, colon, lung and liver cancers [27]. In addition, using microarray data, Ho *et al.* [8] also noticed that an increase in gene expression variance in a disease condition such as cancer is more common than a decrease. Our work confirms these results, extends them to other cancers and increases their reliability by using RNA-seq data and methods based on a more appropriate statistical framework, and rigorously validates them in a simulation study. This increase in the dispersion in gene expression in tumors may reflect the huge variety of genetic perturbations occurring in their development and their polyclonal origin [54]. It may result from a loss of control of gene expression in cancer cells, *e.g.* loss of specificity in signaling cascades, transcriptional activity (cis and trans factors) or post-transcriptional regulation, *e.g.* splicing events or translation inhibition by microRNAs [55, 56]. Whatever its origin, this high variability in gene expression in cancer cells may be considered as a gain of robustness, as defined by Kitano [57]. The increase in the dispersion in the expression of hundreds of genes in tumors may enable the adaptation of the best fitted clone quickly and effectively to any perturbation of the environment. This may explain the resistance to treatment often observed, in particular to treatments that were effective during the first years of application [54]. These genes, whose mean expression does not vary significantly but whose dispersion of expression increases in cancer, form *de facto* a new space for the discovery of potential biomarkers.

**Specificity of the overrepresented functions among DD+ genes.**   Although DD genes are the main focus of interest in our approach, we also identified biological processes enriched among upregulated (S4 File) and downregulated genes (S5 File) in tumors with respect to healthy samples. Among all the previously discussed catabolic GO terms enriched among DD + genes, "mRNA catabolic process" and "protein targeting" are not found to be enriched among DE genes for any dataset while the others are found to be enriched among upregulated genes for less datasets and less significantly (S5 File). Taking into account the higher power of DE gene identification in comparison with DD gene identification, these results reveal that the expression of genes involved in these biological processes is more driven by an increase of dispersion rather than an increase of mean in tumors. Thus, these results highlight the interest in

searching for changes in dispersion, in addition to changes in mean, to yield new insights into tumoral development and cancer treatment efficacy.

## Evaluation of differential dispersion detection performances of the different methods

Based on our simulation study, we demonstrated that GAMLSS and DiffDist outperform all the other evaluated methods, DiPhiSeq is the most robust method and MDSeq must only be applied to lowly DE genes to reliably identify differential dispersion.

**Sensitivity to the presence of a mean fold change.**  We showed that MDSeq tends to falsely identify differential dispersion among highly DE genes. The ability of MDSeq to predict differential dispersion for genes with opposite differential means is indeed poor, with a high level of false positives (S5 Fig). The GLM implemented in MDSeq is based on a reparameterization of the NB distribution, which has the advantage of explicitly modeling the variance of the random variable $Y_{ig}$ describing the read counts but does not allow to directly estimate the dispersion parameter of the classical NB distribution. Under this canonical model, the mean-variance relationship is defined by a quadratic function $\text{Var}(Y_{ig}) = \mu_{ig} + \phi_{ig} \mu_{ig}^2$. Thus, a change in variance may be due only to a change in mean, which explains why MDSeq achieves poor differential dispersion performance among highly DE genes but can still be reliably applied to identify differential dispersion among lowly DE genes, based on a nonsignificant p-value for the difference of mean test and a significant p-value for the difference of variance test (S5 Fig). In contrast, DiPhiSeq, GAMLSS and DiffDist are based on the classical definition of the NB distribution and therefore allows to estimate changes in dispersion. Our evaluations demonstrated that the detection of differential dispersion with DiPhiSeq is not sensitive to the presence of a mean fold change and thus confirmed the claim of the DiPhiSeq authors: their methods effectively handle negatively associated mean and dispersion values [40]. On the contrary, the presence of a mean fold change impacts the performance of GAMLSS and DiffDist but in a lesser extent in comparison with MDSeq, leaving room for methodological improvements. Even if the main interest of searching for DD genes is to identify genes which are not detected by the classical differential expression analysis, *i.e.* non- or lowly DE genes, estimating differences in dispersion irrespectively of mean variation may help avoid misinterpreting a difference in dispersion as a difference in mean, and eventually bring new biological insights [58].

**Overall differential dispersion detection performance.**  The overall much better sensitivity of GAMLSS and DiffDist, especially for sample sizes from 20 to 50 samples per condition, may be explained by their underlying models. The hierarchical Bayesian model of DiffDist indeed enables to share information between genes while estimating dispersion parameters, resulting in more stable estimates. On the contrary, all the other evaluated methods estimate dispersion for each gene independently. Besides, the two other GLM-based methods, *i.e.* MDSeq and DiPhiSeq, are constrained to adopt alternative strategies to estimate differences of dispersion, such as the reparametrization of the NB distribution or the use of M-estimators, respectively. By contrast, the statistical framework of GAMLSS inherently enables the estimation of differences of dispersion and, thus, appears to be advantageous over these two methods.

Levene's test is an alternative to the F-test of equality of variances for data exhibiting departures from the normal distribution. We included in our study this simple statistical test of difference of variances because of the non-normality of the expression of large sets of genes after $\log_2$-transformation (S6 Fig). Surprisingly, we found that this test performed similarly or, in some conditions, even better than MDSeq and DiPhiSeq. Yet, the authors of MDSeq claimed

that their method is more powerful than Levene's test based on simulated datasets where no difference in means and large differences in dipersions were introduced. We hypothesized that our more realistic simulations with moderate to high values of mean fold change, the sensitivity of MDSeq to differences in mean expression and the more conservative FDR-controlling procedure applied to MDSeq outcomes explain our results. Nevertheless, despite a lower sensitivity, the application of these methods to TCGA datasets revealed that MDSeq specifically identified more DD+ genes than Levene's test.

**Specific features.** An advantage of MDSeq and GAMLSS over the other evaluated methods is the inclusion of any additional covariate in their statistical models to prevent some sources of bias from confounding the comparison of interest. Similar to most differential expression analysis methods based on the NB distribution [36, 59], MDSeq implements a GLM that may take into account classical sources of bias, such as batch effects, in the detection of DD genes (see, for example, *LMAN2* expression in the lung adenocarcinoma dataset in S7 Fig) and therefore appears to better handle technical biases. On the contrary, DiPhiSeq and DiffDist do not allow the inclusion of any additional covariate in their respective statistical models. In the case of DiPhiSeq, this limitation is partly mitigated by the use of a Tukey's biweight function that removes any aberrant value regardless of its source, either biological or technical. Moreover, MDSeq also implements a zero-inflated model for which the goal is to control the statistical bias that may be introduced by an excess of null values in the analysis of gene expression data, which is particularly relevant for the analysis of single-cell expression data.

## Gene expression variability at the single-cell level

Without any further specification, gene expression variability usually refers to cell-to-cell variability. Here, we analyzed RNA-seq data of samples composed of thousands of cells, *i.e.* bulk data coming from a population of different individuals. Some studies have demonstrated the limitations of inferences from bulk data regarding gene expression variability [60, 61]. Such approaches are unable to capture cell-to-cell variability and tend to average gene expression [62, 63]. Nevertheless, the estimation of gene expression based on this type of data may still exhibit some variability and provide a snapshot of the expression variability of a gene within populations of cells. We indeed identified a large number of genes with a significant change in dispersion in their expression between healthy and tumor bulk samples from different individuals. Single-cell RNA sequencing technologies have emerged in the last few years, enabling the study of gene expression variability at the cellular level. The application of differential dispersion identification methods to this type of data is promising for a wide range of biological contexts. For example, in the context of cancer, the population of cells composing a tumor may exhibit a high level of gene expression variability, potentially leading to therapeutic failures [64]. However, analyzing the data generated with these techniques faces new methodological issues. Technical null values, or dropouts, are much more present in single-cell RNA-seq data than in bulk RNA-seq data due to the limited amount of mRNA material available at the cellular level [65]. Besides, the high number of samples of single-cell RNA-seq datasets with respect to bulk RNA-seq datasets and the usually lower numbers of expressed genes observed make the read count matrices much more sparse [65] and computation time longer. All these issues should be addressed to reliably apply differential dispersion identification methods to this new type of data.

## Conclusion

We evaluated the performances of four NB-based methods and a statistical test of difference of variances to identify DD genes in RNA-seq data. GAMLSS and DiffDist have very similar

performances and are much more sensitive than the other evaluated methods. Their better performances are mitigated by the erroneous prediction of dispersion $\log_2$-fold change sign for large sets of DD genes, especially for low sample sizes. DiPhiSeq is the most robust method by generating very rare false positive results but lacks power, especially for low sample sizes, while Levene's test and MDSeq performed worst among the evaluated methods. Overall, we recommend the application of either GAMLSS or DiffDist and the validation of the results by DiPhiSeq regarding dispersion $\log_2$-fold change sign.

The application of the differential dispersion approach to gene expression data is relevant to gain knowledge of tumor progression and cancer treatment efficacy. With the emergence of comprehensive RNA-seq datasets, composed of either single-cell or bulk samples in a variety of biological contexts, we believe that the changes in dispersion in gene expression between samples from different conditions of biological interest should now be taken into account. In the classical differential mean expression analysis, it would provide a more realistic model of the data and, in the explicit goal of identifying genes with a differential variance in their expression, it may contribute to gaining new insights into biological processes and eventually to discovering new biomarkers and therapeutic avenues.

## Materials and methods

### RNA-seq dataset simulation

We simulated RNA-seq count datasets using the compcodeR R package [66]. The simulated datasets are composed of 10 000 genes and two sets of samples of equal size corresponding to biological conditions of interest. Read counts for gene $i$ and sample $j$ are generated by random sampling using a negative binomial distribution: $Y_{ij} \sim \mathcal{NB}(\mu_{i\rho(j)}, \phi_{i\rho(j)})$, where $\mu_{i\rho(j)}$ and $\phi_{i\rho(j)}$ are the mean and dispersion parameters, respectively, and $\rho(j)$ denotes the condition of sample $j$ ($\rho(j) \in \{1; 2\}$). The mean $\mu_{i1}$ and dispersion $\phi_{i1}$ values for the first condition are estimated by pairs from real datasets [67, 68]. The mean and dispersion parameters for the second condition were generated by selecting a fraction of the genes to be subjected to a fold change in mean or dispersion.

The dispersion of genes chosen to be differentially dispersed (DD) was determined as $\phi_{i2} = FC_i^\phi \times \phi_{i1}$, where $FC_i^\phi = \delta_i^\phi(FC_i^{\phi,min} + c_i^\phi)$, with $FC_i^{\phi,\min}$ being a predefined minimum fold change, $c_i^\phi$ an extra amount drawn from an exponential distribution of parameter $\lambda = 1$, and $\delta_i^\phi$ which is equally likely to be 1 or −1, so that half the genes have an increase in dispersion and the other half a decrease in dispersion in the second condition. We set $FC_i^{\phi,\min} = 1.5$ to ensure at least a 50% difference in dispersion between the two conditions. The non-DD genes have the same dispersion in both conditions: $\phi_{i2} = \phi_{i1}$.

The mean expression of genes chosen to be differentially expressed (DE) was simulated according to two scenarios:

- Unconstrained mean expression fold changes: Similar to what was done for the dispersion parameter of DD genes, $\mu_{i2} = FC_i^\mu \times \mu_{i1}$, where $FC_i^\mu = \delta_i^\mu(FC_i^{\mu,\min} + c_i^\mu)$. We set the mean minimum to 50%, *i.e.* $FC_i^{\mu,\min} = 1.5$, $c_i^\mu$ was drawn from an exponential distribution of parameter $\lambda = 1$ and $\delta_i^\mu$ was equally likely to be 1 or −1. Since the evaluation of differential mean expression detection performance is not the main goal of our approach, we simulated non-DE genes in a more realistic way than having the same mean expression value for both conditions. Instead, we allowed slight fold changes by random sampling using uniform distributions: $FC_i^\mu \sim \mathcal{U}(1, FC_i^{\mu,\min})$, where the maximum value corresponds to the minimum value of fold change for highly DE genes. These genes are therefore called lowly DE genes rather than non-DE genes.

- Moderated mean expression fold changes: Since the purpose of these datasets is only to assess differential dispersion detection performance for lowly DE genes, the distinction between DE and non-DE genes is not required. For all the genes of these datasets, mean fold changes were introduced using uniform distributions: $FC_i^\mu \sim \mathcal{U}(1, FC_i^{\max})$, where $FC_i^{\max} \in \{1.1;\ 1.2;\ 1.3;\ 1.4;\ 1.5\}$.

For all the simulated datasets, the same fractions of DD genes (or non-DD genes) among highly DE genes and lowly DE genes (for the first set of simulations) were ensured, as well as the same fractions of DD genes with an increase in dispersion in the second condition (DD+) among upregulated genes (DE+) and downregulated genes (DE-) in the second condition. Thus, simulated datasets are composed of 50% DD genes and 50% non-DD genes and 50% highly DE genes and 50% lowly DE genes for the first set of simulations.

For the sake of realism, we introduced outliers with very high counts in all the simulated datasets since Li *et al.* [40] showed that they may have a dramatic effect on differential dispersion detection. Following the recommendations of Soneson and Delorenzi [45], we multiplied one read count by a value from 5 to 10 for 10% of the genes.

## RNA-seq data preprocessing

Before applying differential dispersion detection methods, classical RNA-seq data preprocessing steps were applied to all the simulated and TCGA datasets. First, read counts were normalized by the Trimmed Mean of M-values method [69, 70]. Lowly expressed genes were then independently filtered out using a threshold of 1 count per million [71].

## Differential dispersion detection

Levene's test [47] was applied to the normalized read counts after $\log_2$-transformation and the addition of a pseudocount of 1 to avoid null values. Genes with a p-value lower than 0.05 after the correction by the Benjamini-Hochberg procedure [72] were considered as differentially dispersed.

The outlier removal function was applied with the minimum sample size lowered to 1 before applying the MDSeq [39] main function to all the simulated and TCGA datasets. Batch effects were handled when analyzing TCGA datasets by supplying the sequencing runs that generated the RNA-seq samples, when available, as a covariate in the GLM for both the identification of DD genes and the identification of DE genes. A fold change threshold of 1 was used to identify both DE genes and DD genes. The default values were used for the other parameters for both the outlier removal and MDSeq functions. The p-values for both differential mean and differential dispersion statistical tests were corrected by the Benjamini-Yekutieli procedure [73] to control the FDR as recommended by the authors of MDSeq.

DiPhiSeq [40] was applied to all the simulated and TCGA datasets with the default values for all the parameters, in particular the *c* parameter of Tukey's biweight function set to 4 for both the mean and the dispersion estimation since the authors of DiPhiSeq found that this value enables robust parameter estimations [40]. The p-values for both differential mean and differential dispersion statistical tests were corrected by the Benjamini-Hochberg procedure [72] to control the FDR as recommended by the authors of DiPhiSeq.

We applied the approach proposed by de Jong *et al.* [41] of using a GAMLSS to identify DD genes by transposing the analysis method presented in the following GitHub repository: https://github.com/Vityay/ExpVarQuant.

DiffDist [42] was applied to all the simulated and TCGA datasets with the default values for all the parameters. The tail probabilities for both differential mean and differential dispersion

statistical were corrected by the Benjamini-Hochberg procedure [72] to control the FDR as recommended by the authors of DiffDist.

## Performance evaluation

The performances of differential dispersion detection methods were evaluated based on the fold changes of dispersion of expression introduced in the simulated datasets. The genes that were simulated to be DD are the positive group for the differential dispersion performance evaluation. The genes that were simulated to be non-DD are the corresponding negative group. For all the evaluated methods, a p-value for the differential dispersion or differential variance statistical test lower than 0.05 after the application of the appropriate FDR-controlling procedure was used to define a positive detection. The comparisons with the positive group enabled us to count true positive and false positive results for differential dispersion. Similarly, true negative and false negative results were identified thanks to a corrected p-value of differential dispersion statistical test higher than 0.05 and the comparisons with the negative group. The sensitivity (or true positive rate), the false discovery rate and the area under the ROC curve were then computed based on these four categories of results.

## Gene Ontology term enrichment analysis across datasets

For each TCGA dataset, genes of interest, *e.g.* DD+ genes among non-DE genes, were identified and Gene Ontology (GO) term enrichment analysis was performed using the Biological Processes (BP) ontology. Enriched GO terms were identified thanks to a hypergeometric test p-value after FDR control lower than 0.05 using the enrichGO function of the clusterProfiler R package [74]. The list of enriched GO terms was then reduced by gathering terms whose semantic similarity exceeded a threshold value. To do so, clusters of closely related GO terms were generated through the relevance method [75] to compute semantic similarity between GO terms. A high similarity threshold (0.8) was used to gather only closely related GO terms into clusters. The GO term whose p-value is the lowest among a cluster was then chosen to represent the entire cluster.

To facilitate comparisons across datasets, closely related GO terms were searched for among the previously simplified enriched GO term lists originating from each dataset. The similarity of all GO term pairs was calculated with the relevance method. These similarity values were then used to perform hierarchical clustering and gather closely related GO terms by using a conservative threshold value (0.8). For each cluster of closely related GO terms, the hierarchical structure of the BP ontology was used to identify a generic term common to all the GO terms. This common generic GO term was subsequently used as the representative term for the entire cluster, and its enrichment p-value was retrieved for each TCGA dataset containing an enriched GO term in the cluster.

## Supporting information

**S1 Fig. Correction of MDSeq p-values for differential dispersion by the Benjamini-Hochberg and Benjamini-Yekutieli FDR-controlling procedures.** P-values obtained with MDSeq for the detection of differential dispersion in gene expression data from a simulated dataset composed of highly and lowly differentially expressed genes between two populations of 50 samples. P-values were corrected by the Benjamini-Hochberg (BH) and Benjamini-Yekutieli (BY) procedures for (A) the entire set of genes and (B) the lowly DE genes only. The red dotted lines represent a p-value threshold value of 0.05.
(PDF)

**S2 Fig. Ability to detect differential dispersion for lowly differentially expressed genes.**
False discovery rate (FDR) and true positive rate (TPR) of Levene's test, MDSeq, DiPhiSeq, GAMLSS and DiffDist for differential dispersion detection in simulated datasets composed of lowly differentially expressed genes between two sample populations of equal size. The Benjamini-Hochberg procedure was applied to control the FDR for all the methods. The performances were assessed using 10 replicates of simulated datasets per parameter setting.
(PDF)

**S3 Fig. Differential dispersion performance with MDSeq after filtering highly DE genes using different fold change threshold values.** (A) False discovery rate (FDR) and (B) true positive rate (TPR) values obtained with simulated datasets composed of two sample populations of equal size (panels on the horizontal axis).
(PDF)

**S4 Fig. Agreement between dispersion $\log_2$-fold change signs predicted by GAMLSS or DiffDist and those predicted by DiPhiSeq.** Counts of differentially dispersed (DD) genes specifically identified by (A) GAMLSS or (B) DiffDist with a sign of dispersion $\log_2$-fold change consistent or inconsistent with the one predicted by DiPhiSeq for all the analyzed TCGA datasets. Percentages of the two defined categories of genes over the entire sets of specifically identified DD genes are indicated at the middle of the corresponding bars.
(PDF)

**S5 Fig. Reliable identification of differentially dispersed genes among lowly differentially expressed genes with MDSeq.** Identification of differentially dispersed genes based on (A) a significant p-value for the difference of variance test or (B) a nonsignificant p-value for the difference of mean test and a significant p-value for the difference of variance test. True mean and dispersion $\log_2$-fold changes (left panels) and estimated mean and variance $\log_2$-fold changes with MDSeq (right panels) of a simulated dataset composed of highly and lowly differentially expressed genes between two populations of 50 samples are displayed. Colours represent the results of the identification of differential dispersion by MDSeq using a $\log_2$-fold change threshold of 0. The red dotted line is the $y = x$ diagonal.
(EPS)

**S6 Fig. Normality of gene expression in simulated datasets composed of lowly DE genes.**
The Shapiro-Wilk test was computed for each gene after $\log_2$-transformation of the normalized read counts. P-values were corrected by the Benjamini-Hochberg procedure and displayed stratified by the sample size. Counts of adjusted p-values lower than 0.05 and adjusted p-values greater than 0.05 were averaged over 10 replicates of simulated datasets and percentages are indicated at the middle of the corresponding bars.
(PDF)

**S7 Fig. Batch effect handling by a covariate in the generalized linear model implemented by MDSeq.** Expression values of the *LMAN2* gene (lectin, mannose binding 2, ENSG00000169223) based on the TCGA dataset composed of samples from patients with lung adenocarcinoma (TCGA-LUAD) for whom both a tumor sample and a normal sample are available. Data are clustered according to sequencing batch. In batches 0946, 1107, 1206, A277 and A278, which enabled the sequencing of only tumor samples, the dispersion of LMAN2 expression increased with respect to the other batches composed of samples from both conditions. Corrected p-values obtained with the five evaluated methods are listed below. MDSeq without the integration of batch effect by a blocking factor in the generalized linear model (GLM): $2.14 \cdot 10^{-4}$, MDSeq with the integration of batch effect by a blocking factor in the GLM:

$1.1410^{-1}$, Levene's test: $1.9510^{-3}$, DiPhiSeq: $9.0810^{-5}$, GAMLSS: $2.9410^{-8}$, DiffDist: $1.4810^{-4}$.
(PDF)

**S1 File. Differentially dispersed genes correctly identified by the evaluated methods among lowly differentially expressed genes.** (A) Intersections of sets of differentially dispersed (DD) genes correctly identified by Levene's test, MDSeq, DiPhiSeq, GAMLSS and DiffDist. (B): Correctness of dispersion $\log_2$-fold change sign of DD genes correctly identified by the different methods. (C) Real mean and dispersion $\log_2$-fold changes and estimated dispersion $\log_2$-fold changes of DD genes correctly identified by GAMLSS and DiffDist. (D) Correctness of dispersion $\log_2$-fold change signs according to Levene's test, MDSeq and DiPhiSeq for DD genes correctly identified by GAMLSS and DiffDist with incorrect $\log_2$-fold change sign. Simulated datasets are composed of two populations of 20, 30, 40 or 100 samples and lowly differentially expressed genes have a mean fold change of expression between 1 and 1.5. All counts are averaged over 10 replicates of simulated datasets.
(PDF)

**S2 File. Overdispersed genes in tumors identified by the evaluated methods among non-differentially expressed genes.** Intersections of sets of overdispersed genes in tumors (DD+) identified by Levene's test, MDSeq, DiPhiSeq, GAMLSS and DiffDist among non-differentially expressed genes for the following TCGA datasets: breast invasive carcinoma (TCGA-BRCA), colon adenocarcinoma (TCGA-COAD), head and neck squamous cell carcinoma (TCGA-HNSC), liver hepatocellular carcinoma (TCGA-LIHC), lung adenocarcinoma (TCGA-LUAD), lung squamous cell carcinoma (TCGA-LUSC), prostate adenocarcinoma (TCGA-PRAD), and thyroid carcinoma (TCGA-THCA). Non-differentially expressed genes were identified by MDSeq.
(PDF)

**S3 File. Enriched GO terms among overdispersed genes in tumors identified by the evaluated methods.** Full list of representative enriched Gene Ontology (GO) terms among overdispersed genes in tumors (DD+) among non-differentially expressed (non-DE) genes, ordered first by the number of datasets for which they are enriched (decreasing order) and second by the mean p-values of enrichment across all datasets (increasing order). Non-DE genes were identified using MDSeq, and DD+ genes were identified among non-differentially expressed genes by at least one of the evaluated methods, *i.e.* Levene's test, MDSeq, DiPhiSeq, GAMLSS and DiffDist.
(PDF)

**S4 File. Enriched GO terms among upregulated genes in tumors for each TCGA dataset.** Full list of representative enriched Gene Ontology (GO) terms among upregulated genes in tumors, ordered first by the number of datasets for which they are enriched (decreasing order) and second by the mean p-values of enrichment across all datasets (increasing order). Upregulated genes were identified by MDSeq using a mean fold change threshold of 1.
(PDF)

**S5 File. Enriched GO terms among downregulated genes in tumors for each TCGA dataset.** Full list of representative enriched Gene Ontology (GO) terms among downregulated genes in tumors, ordered first by the number of datasets for which they are enriched (decreasing order) and second by the mean p-values of enrichment across all datasets (increasing order). Downregulated genes were identified by MDSeq using a mean fold change threshold of 1.
(PDF)

## Acknowledgments

We thank Adeline Leclercq-Samson and Christophe Battail for their helpful recommendations.

## Author Contributions

**Conceptualization:** Christophe Le Priol, Chloé-Agathe Azencott, Xavier Gidrol.

**Formal analysis:** Christophe Le Priol.

**Funding acquisition:** Xavier Gidrol.

**Investigation:** Christophe Le Priol, Chloé-Agathe Azencott, Xavier Gidrol.

**Methodology:** Christophe Le Priol, Chloé-Agathe Azencott.

**Software:** Christophe Le Priol.

**Supervision:** Chloé-Agathe Azencott, Xavier Gidrol.

**Validation:** Christophe Le Priol.

**Visualization:** Christophe Le Priol.

**Writing – original draft:** Christophe Le Priol.

**Writing – review & editing:** Chloé-Agathe Azencott, Xavier Gidrol.

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
