## [Decision Letter · Decision Letter 0]

7 Oct 2022

Dear Mr Le Priol,

Thank you very much for submitting your manuscript "Detection of genes with differential expression dispersion unravels the role of autophagy in cancer progression" for consideration at PLOS Computational Biology.

As with all papers reviewed by the journal, your manuscript was reviewed by members of the editorial board and by several independent reviewers. In light of the reviews (below this email), we would like to invite the resubmission of a significantly-revised version that takes into account the reviewers' comments.

Please address concerns raised by all reviewers, but please also pay special attention to the concern raised by Reviewer 1 regarding the motivation of this study. (This concern is also echoed by Reviewer 3, who questions the process for selection of the two evaluated methods.)

We cannot make any decision about publication until we have seen the revised manuscript and your response to the reviewers' comments. Your revised manuscript is also likely to be sent to reviewers for further evaluation.

Sincerely,

Saurabh Sinha

Guest Editor

PLOS Computational Biology

Ilya Ioshikhes

Section Editor

PLOS Computational Biology

Please address concerns raised by all reviewers, but please also pay special attention to the concern raised by Reviewer 1 regarding the motivation of this study. (This concern is also echoed by Reviewer 3, who questions the process for selection of the two evaluated methods.)

Reviewer's Responses to Questions

**Comments to the Authors:**

Reviewer #1: The authors review two methods for finding differentially variable genes, MDSeq and DiPhiSeq. They then apply both methods to TCGA data to find that differentially variable genes in cancer are enriched in some interesting Gene Ontology categories.

The work is very thorough but I'm not sure why the authors felt that MDSeq and DiPhiSeq were important to evaluate. Have these two methods made such an impact on the subfield that they deserve to be studied and compared? Have there been no other methods proposed since then that also deserve to included in the comparison? I'm not convinced that these two methods are important to study.

The study itself also does not include any simple tests for differential variance. For completeness I would be interested in seeing how something like a simple permutation approach using a statistic with power toward differential variances instead of differential expression would perform.

The authors apply these methods to gain into cancer. However, I'm not sure this gives me any additional insight about the methods comparison. The biological results generate interesting hypotheses, but there is limited validation of the results and in any case the purpose of this report doesn't seem to be to produce new biological insights.

Reviewer #2: I recommend that the manuscript be considered for publications after minor revisions.

Here, the authors compare and contrast 2 methods- DiPhiSeq and MDSeq for their utility of detecting differentially dispersed (DD) genes in bulk RNA-Seq data. Through their usage on simulated data and TCGA cancer RNA-Seq data, the authors have shown that DiPhiSeq provides a better performance in detecting DD genes despite smaller sample sizes and effect sizes. The study provides a detailed insight into the accuracy, sensitivity and specificity, computational resources and preprocessing requirements across both methods. This is a timely and valuable study for DD estimation as bulk RNA-Seq datasets are generally analyzed using differential expression methods, but variance and dispersion between genes is rarely compared.

Comments (in order of priority):

1. Further clarification is needed on the choice of the methods used for comparison- DiPhiSeq and MDSeq. What kind of literature search was involved in making sure that these are the only methods available for calculating differential dispersion/distributions.

2. How do different preprocessing methods (normalization, filtering etc.) impact the downstream results from the two tools? For example, in this case, TMM was used to normalize. If other methods of normalization or different thresholds for filtering were used, would you expect the methods to behave differently?

3. In lines 119-125, you mention the different methods used for adjustment for multiple testing. How do the methods compare if the same FDR adjustment method was used- BH/BY/Bonferroni? Please elaborate this sentence- "We note, however, that in our evaluation (see S2 Fig), the Benjamini-Hochberg procedure was not sufficient to control for FDR".

4. It is important to provide some recommendations for the usage of either of the two methods. Do you suggest that they be used in conjunction to DE? Or consecutively?

5. In lines 140-141, MDSeq is suggested to be used in lowly DE genes. You provide fold change thresholds based on your simulation studies. How should these be interpreted for tissues or conditions where gene expression is relatively low or the effects are subtle?

6. In Figure 3, "True DD" term is used. How are these classified? Further clarification is needed on the 'gold standard' for DD.

7. In lines 220-221, the number of non-DE genes are substantially different between the two methods. These numbers invariably impact the number of DD tests run. Please elaborate further on how this difference in number of non-DE genes impact the number of DD genes classified by the models.

8. I suggest that if you plan on including discussion on single-cell datasets, some validation is required, especially for the claim that MDSeq may be more useful.

9. The results from the TCGA DD analysis are extremely useful not only to establish the utility of the tools but also interpreting DD genes as having a role in robustness and regulation. The GO analysis and the interpretation seems very speculative at the moment. The discussion includes hand-picking a few pathways that showed up from the GO analysis. I suggest that the the biological findings be enhanced by analyzing a treatment dataset for one of the cancer types chosen from TCGA to further show how a potential 'therapeutic' can regulate the DD genes in a way opposite to that from cancer etiology.

Reviewer #3: In this manuscript, Le Priol et al. evaluated two different tools to analyze differentially variant genes in large genomic datasets, and found the relevance of these genes in autophagy in cancer. Authors performed a thorough investigation of MDSeq, and DiPhiSeq, and identified parameter/parameter sets, minimum number of sample size required for these tools to be efficient. Then, they applied the tools on one large public repository, TCGA, to examine the biological functions of the genes that are differentially variant.

Overall, the manuscript is well-drafted, and would be of interest to the general readers. Overall, I believe the manuscript is well-suited as a tool/resource.

**Have the authors made all data and (if applicable) computational code underlying the findings in their manuscript fully available?**

Reviewer #1: Yes

Reviewer #2: Yes

Reviewer #3: Yes

PLOS authors have the option to publish the peer review history of their article (what does this mean?). If published, this will include your full peer review and any attached files.

Reviewer #1: No

Reviewer #2: No

Reviewer #3: No
---

## [Decision Letter · Decision Letter 1]

9 Feb 2023

Dear Mr Le Priol,

We are pleased to inform you that your manuscript 'Detection of genes with differential expression dispersion unravels the role of autophagy in cancer progression' has been provisionally accepted for publication in PLOS Computational Biology.

Best regards,

Saurabh Sinha

Guest Editor

PLOS Computational Biology

Ilya Ioshikhes

Section Editor

PLOS Computational Biology

Reviewer's Responses to Questions

**Comments to the Authors:**

Reviewer #1: The editors have sufficiently addressed my comments.

**Have the authors made all data and (if applicable) computational code underlying the findings in their manuscript fully available?**

Reviewer #1: None

PLOS authors have the option to publish the peer review history of their article (what does this mean?). If published, this will include your full peer review and any attached files.

Reviewer #1: No

---

## [Editor Report · Acceptance letter]

23 Feb 2023

PCOMPBIOL-D-22-01003R1 

Detection of genes with differential expression dispersion unravels the role of autophagy in cancer progression

Dear Dr Le Priol,

I am pleased to inform you that your manuscript has been formally accepted for publication in PLOS Computational Biology. Your manuscript is now with our production department and you will be notified of the publication date in due course.

With kind regards,

Anita Estes
